# Research into a Consistency Cooperative Control Method for Attitude Orbit Coupling of SAR Satellite Formations under Communication Constraints

**Bing Hua \*, Shengjun Zhong** **, Yunhua Wu and Zhiming Chen** 

Department of Aerospace Control Engineering, Aerospace Academy Department, Nanjing University of Aeronautics and Astronautics, Nanjing 211100, China
\* Correspondence: huabing@nuaa.edu.cn; Tel.: +86-138-1380-2610

**Abstract:** To solve the problem of communication delays and topology changes and achieve a high-precision formation configuration control index in the flight control of SAR (synthetic aperture radar) satellite formations, this paper studies the consistency cooperative control of attitude orbit coupling for SAR satellite formations under communication constraints. First, this paper establishes the relative motion state error equation for satellite formations. Considering the capacity constraints for communication and actuators, this paper designs a hierarchical saturated consistency cooperative controller for attitude orbit coupling. Second, this paper uses the Lyapunov direct method to prove the stability of the designed consistent cooperative controller under uncertain space disturbances. Finally, this paper simulates and verifies the designed controller. The results show that the hierarchical saturation consistency cooperative controller designed in this paper can meet the requirements of configuration maintenance accuracy for SAR satellite formation for Earth target observation missions.

**Keywords:** SAR satellite formation; communication delay; topology; attitude orbit coupling; consistency cooperative control



## 1. Introduction

SAR satellite formation can achieve high-efficiency ground moving target detection performance through cooperative work and imaging processing of multiple spaceborne radars. When a SAR satellite formation carries out the ground moving target detection task, high-precision configuration position control and attitude tracking control is required. The high-precision control of satellites is conducive to the completion of satellite missions. In some current studies, the high-precision formation satellite flight control is mainly based on the consistency collaborative control [1–11].

Consistency collaborative control means that the movement of the actuator control system tends to be consistent with the exchange of fusion information, continuous feedback, and exchange of information among all independent agents in the multiagent system. In the formation flight control of SAR satellites, the relationship between information exchanges with SAR satellites and the performance of the formation control system can be obtained through consistency cooperative control. This is beneficial for the design of the formation flight control system and makes the control form of the formation system more general. Reference [7] designed a robust cooperative control law for the hovering target of spacecraft formation by introducing a consistent control theory about the condition that the electromagnetic interaction model and dynamic equations are uncertain. Reference [8] proposed an adaptive cooperative collision avoidance control law with strong robustness based on consistency cooperative control and considering various constraints for flight formation. Reference [9] designed a cooperative control law for the optimal orbital manoeuvre of a spacecraft based on the first-order consistency control theory about the cooperative flight manoeuvre mission of a spacecraft in formation. Reference [10] designed a fault-tolerant

controller based on sliding mode cooperative control and showed that the proposed control method tolerates the actuators' faults and controls the satellite's attitude while desaturating the reaction wheels. Reference [11] proposed a coordinated control to fulfil these constraints for impulsive formation maintenance tailored to distributed synthetic aperture radar.

When the SAR satellite formation carries out consistent cooperative control, the member satellites must interact with each other to determine their own control behaviour. However, in the process of information exchange, on the one hand, considering the constraints of distance and signal receiving equipment, and on the other hand, considering factors such as satellite formation manoeuvre, obstacle avoidance, fault, and so on, it is inevitable that topology switching, communication delay, and other problems will occur. Those factors will reduce the control accuracy of the whole satellite formation control system and affect the stability of the control system. To solve the above problems, Reference [12] designed a terminal sliding mode attitude cooperative controller and proposed a design method based on a spacecraft formation cooperative controller based on an exponential logarithmic sliding mode surface for the situation of communication delays and signal quantization between satellite formations. Reference [13] designed a robust controller with good steady-state performance for the case of communication delays and signal quantization between satellite formations. Reference [14] put forward an optimal control method with guaranteed-performance and switching topologies for the formation achievement problem of swarm systems. Reference [15] designed a distributed model predictive control algorithm considering multiple constraints in order to realize trajectory tracking and formation keeping of a multiUAV system on the premise of meeting the above constraints. In Reference [16], time delayed estimation was used to estimate parameter uncertainty caused by external disturbances in satellite dynamics. Then, the time delay estimation output was combined with a robust TSM controller based on interval type-2 fuzzy logic. In Reference [17], an adaptive continuous robust controller was designed to compensate for the influence of model uncertainty on satellite formation when the time delay is uncertain.

The information exchanges among the members of the SAR satellite formation depend on wireless communication, while the traditional SAR satellite formation cooperative control technology often ignores some unfavourable factors, such as the communication delay factor. Therefore, this paper studies the cooperative control of SAR satellite formation flight under multiple constraints. Considering the constraints of actuator output capability and uncertain space disturbance, this paper designs a hierarchical saturated consistency coordinated control of attitude orbit coupling in order to solve the problem of communication delays in the configuration for keeping control of SAR satellite formation. In the end, this paper proves the stability of the controller.

## 2. Attitude Orbit Coupling Model of SAR Satellite Formation

This paper selects the coordinate diagram of the SAR satellite formation system [18,19], as shown in Figure 1. In the figure, $OXYZ$ is the geocentric inertial coordinate system, in which axis $X$ points to the vernal equinox, axis $Z$ is the Earth's rotation axis, and axis $Y$ meets the right-hand coordinate system with axis $X$ and axis $Z$; $oxyz$ is the relative motion coordinate system, in which axis $x$ is the satellite geocentric distance direction, axis $y$ is the satellite speed direction, and axis $z$ meets the right-hand coordinate system with axis $x$ and axis $y$; $o_i x_i y_i z_i$ is the $i$ th satellite body coordinate system, in which the three-axis direction is consistent with the satellite inertia axis, $o_i$ is the centroid for the $i$ th satellite of the formation, $\boldsymbol{r}$ is the position vector from the reference satellite centroid of the Earth centre, $\boldsymbol{r}_i$ is the position vector from the centroid for the $i$ th satellite of the formation to the reference satellite, and $\boldsymbol{r}_{f_i}$ is the position vector from the centroid for the $i$ th satellite of the formation to the Earth centre. In the paper, the '$i$' subscript at the lower right of the satellite state parameters is uniformly expressed as the state parameters of the $i$ th satellite.

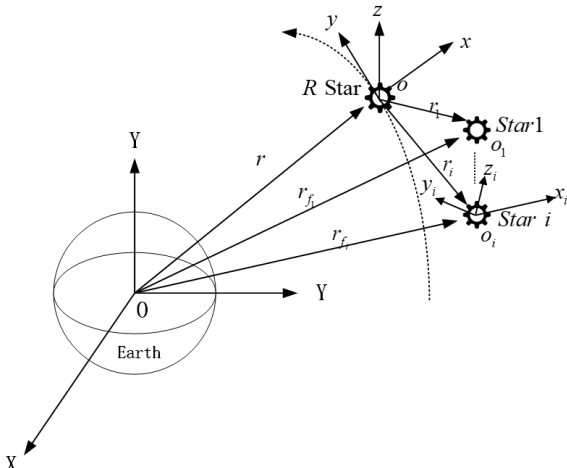

**Figure 1.** Schematic diagram for the SAR satellite formation system coordinates.

The dynamic and kinematic equations of the *i*-th member satellite represent any satellite in the formation. Let the position and velocity vectors of the satellite formation relative to the reference satellite in the equatorial inertial coordinate system be $r_i$ and $v_i$, respectively. Then, the orbit dynamics equation for the satellite is given by

$$\ddot{r}_i = f_g(r_i) + f_j(r_i) + f_c(r_i) \tag{1}$$

where $f_g(r_i)$, $f_j(r_i)$, and $f_c(r_i)$ are the gravitational acceleration, perturbation acceleration, and control acceleration of the Earth centre, respectively.

In the coordinate system of the *i* th satellite, the kinematic equation is expressed as

$$\dot{r}_i = v_i - \omega_i \times r_i \tag{2}$$

where $\omega_i$ is the representation of satellite velocity in the volume coordinate system.

To show the influence of attitude to orbit control, the dynamic equation for the attitude orbit coupling with the *i* th satellite can be obtained by projecting the orbit dynamic equation in the inertial coordinate system onto the satellite body coordinate system:

$$m\ddot{r}_i = -2m\omega_i \times \dot{r}_i - m\dot{\omega}_i \times r_i - m\omega_i \times (\omega_i \times r_i) + f + f_d \tag{3}$$

where $m$ is the mass of the satellite, $r$ is the position vector of the satellite relative to the Earth centre in the volume coordinate system, $\omega_i = [\omega_x, \omega_y, \omega_z]^T$ is the angular velocity of the satellite, $f$ is the control force, and $f_d$ is the J2 perturbation and other interference forces.

By substituting Equations (1) and (2) into Equation (3), the orbit dynamics equation containing the satellite rotation angular velocity is obtained as follows:

$$m\dot{v}_i = -m\omega_i^\times v_i + f + f_d \tag{4}$$

$$\omega^\times = \begin{bmatrix} 0 & -\omega_3 & \omega_2 \\ \omega_3 & 0 & -\omega_1 \\ -\omega_2 & \omega_1 & 0 \end{bmatrix} \tag{5}$$

The attitude dynamics and motion equations of any satellite in the satellite formation are given by

$$J_i\dot{\omega}_i = -\omega_i^\times J_i\omega_i + \tau_i + \tau_{id} \tag{6}$$

$$\dot{Q}_i = \frac{1}{2}\begin{bmatrix} -q_i^T \\ q_i^\times + q_{i0}I \end{bmatrix}\omega_i \tag{7}$$

where $J$ is the satellite moment of inertia, $\omega_i$ is the rotational angular velocity, $\tau_i$ is the control torque, and $\tau_{id}$ is the interference torque.

Assuming that the reference satellite for the SAR satellite formation always exists on an ideal state, it is only necessary to carry out cooperative control on the slave satellite. This will prevent the entire formation satellite from being in an unstable state at the same time. According to the desired configuration and attitude constraints on the formation, the reference is set to the satellite target. If we set the orbit position and velocity of the reference star relative to the inertial system as $r_d$ and $v_d$ in the slave star coordinate system and the slave star position as $r$ and $v$ in the body coordinate system, then the position error $r_e$ and velocity error $v_e$ are given by

$$\begin{aligned} v_e &= v - v_d \\ r_e &= r - r_d \end{aligned} \tag{8}$$

The attitude angle and angular velocity of the reference star are set as $Q_d$ and $\omega_d$, and the attitude angle and angular velocity of the slave star are set as $Q$ and $\omega$. Then, the quaternion error and attitude angular velocity error are given by

$$Q_e = Q^{-1} \otimes Q_d = \begin{bmatrix} q_0 \\ -q \end{bmatrix} \otimes Q_d \tag{9}$$

$$Q_i \otimes Q_j = \begin{bmatrix} q_{i0}q_{j0} - q_i^T q_j \\ q_{i0}q_j + q_{j0}q_i + q_i^\times q_j \end{bmatrix} \tag{10}$$

$$\omega_e = \omega - T(Q_e)\omega_d \tag{11}$$

where the rotation matrix from the reference star coordinate system to the slave star coordinate system is given by

$$T(Q_e) = (q_{e0}^2 - q_e^T q_e)I_3 + 2q_e q_e^T - 2q_e q_e^\times \tag{12}$$

$$\|T(Q_e)\| = 1 \tag{13}$$

$$\dot{T}(Q_e) = -\omega_e^\times T(Q_e) \tag{14}$$

In the paper, the '*ie*' subscript at the lower right of the satellite state parameters is uniformly expressed as the state error parameters of the *i* th satellite. By substituting the above error Equations (8)–(14) into the kinematics and dynamics Equations (4)–(7), the following attitude and orbit kinematics and dynamics equations for formation satellite errors can be obtained:

$$\begin{cases} m\dot{v}_{ie} = -m\omega^\times v_{ie} + f + f_d \\ J\dot{\omega}_{ie} = -\omega_i^\times J\omega_i + J\omega_i^\times T(Q_{ie})\omega_{id} + \tau + \tau_d \\ \dot{r}_{ie} = v_{ie} - \omega_i^\times r_{ie} \\ \dot{Q}_{ie} = \frac{1}{2}\begin{bmatrix} -q_{ie}^T \\ q_{ie}^\times + q_{ie0}I_3 \end{bmatrix}\omega_{ie} \end{cases} \tag{15}$$

## 3. Design of the Cooperative Controller

### 3.1. Communication Graph Theory

In the face of complex satellite formation missions, it is inevitable that the communication between satellite formations will be affected, such as communication delay and topology switching. To express the communication problem in formation flight control in the form of an abstract graph, graph theory is usually used to describe the problem. The communication topology between satellite formations can be described by using directed graphs and undirected graphs [20]. The digraph and undirected graph are shown in Figure 2.

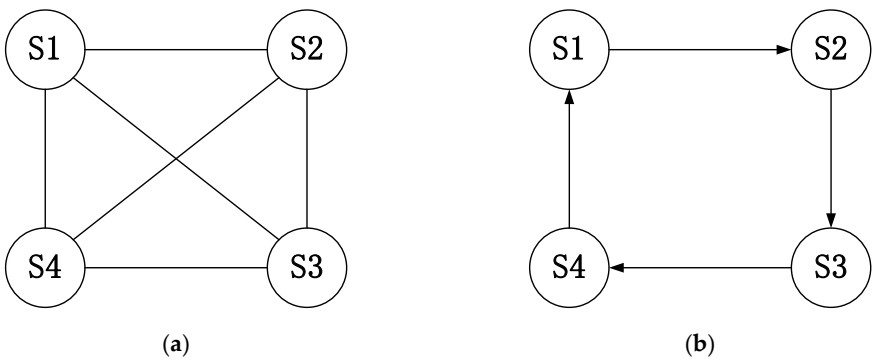

**Figure 2.** Communication graphs for SAR satellite formation. (**a**) The digraph graph; (**b**) the undirected graph.

This can be represented by a triple $(o_i, \varsigma_i, C)$, where $o_i = \{o_1, o_2, \ldots, o_i\}$ is called a node set, $\varsigma_i \subseteq o_i \times o_i$ is called an edge set, and $\boldsymbol{G} = [g_{ij}] \in R^{n \times n}$ is a weighted adjacency matrix. For the nodes in the whole structure, side $(i, j)$ means that satellite $i$ can accept the satellite information and meet $(i, j) \in \varsigma_i$. For a directed graph, if two nodes are connected by edges, the data transmission between the two nodes is unidirectional: that is, if $(i, j) \in \varsigma_i$, then $(j, i) \notin \varsigma_i$. For an undirected graph, if two nodes are connected by edges, the data transmission between the two nodes is bidirectional: that is, if $(i, j) \in \varsigma_i$, then $(i, j) \in \varsigma_i$. For the weighted adjacency matrix $\boldsymbol{G}$ describing the node relationship, $g_{ij} = 1$ means that node $j$ can receive the information from node $i$, and $g_{ij} = 0$ means that node $j$ cannot receive the information from node $i$.

### 3.2. Consistency Collaborative Control

When there are reference satellites outside, the member satellites in the formation should not only meet the relative control between member satellites but also meet the absolute control between member satellites and external reference satellites. This requires the motions between member satellites and member satellites to agree. This also requires the motions between member satellites and reference satellites to agree. This chapter designs the control input for the formation system and the state feedback on the system communication topology. The controller algorithm [21] $u_i$ is expressed as follows:

$$u_i = -\sum_{j=1}^{n} g_{ij}\big[(x_i - x_j) + k(\gamma_i - \gamma_j)\big] \tag{16}$$

where $g_{ij}$ is the $j$ column term in row $i$ of the communication topology matrix, $x_i, \gamma_i$ are the state information for the $i$ th satellite, and $k$ is a constant greater than zero.

When the communication between satellite formations changes, the communication topology $\boldsymbol{G}_{ij}$ will change accordingly. Therefore, considering the constraint on communication topology, this chapter discusses the design of a controller that can make the state variables $x_i$ and $\gamma_i$ of each member satellite tend to be consistent when the control time approaches infinity. When the information on each satellite system is connected, the state information variables for other satellite systems can be obtained. At this time, the state variables for each satellite system tend to be consistent.

### 3.3. Design of a Consistent Cooperative Controller

When the SAR satellite formation system performs consistent cooperative control, the absolute control items of the member satellite and the reference satellite in the orbit and attitude cooperative controller design are expressed as follows:

$$\boldsymbol{u}_{1i} = -\sum_{j=1}^{n} g_{ij}[(\boldsymbol{r}_i - \boldsymbol{r}_d) + k(\boldsymbol{v}_i - \boldsymbol{v}_d)] \tag{17}$$

$$u_{2i} = -\sum_{j=1}^{n} g_{ij}[(q_i - q_d) + k(\omega_i - \omega_d)] \tag{18}$$

where $r_i, v_i, q_i, \omega_i$ are the position and attitude information of the $i$ th satellite.

In practice, the information received by member satellite $i$ from member satellite $j$ is not the current real-time information on member satellite $j$ but the status information on $T_{ij}$ seconds ago. Therefore, the controller needs to consider the communication delay. That is, the status information for $x_j(t - T_{ij})$ is used, not the status information for $x_j(t)$. In the orbit and attitude co-controller [22], the relative control items for member satellites are designed as follows:

$$u_{1i} = -\sum_{j=1}^{n} g_{ij}\left[(r_i - r_j(t - T_{ij})) + k(v_i - v_j(t - T_{ij}))\right] \tag{19}$$

$$u_{2i} = -\sum_{j=1}^{n} g_{ij}\left[(q_i - q_j(t - T_{ij})) + k(\omega_i - \omega_j(t - T_{ij}))\right] \tag{20}$$

Therefore, the track and attitude controllers are designed to ensure that the control items meet some conditions: $\lim_{t\to\infty} Q_{ie} = [1\ 0\ 0]^T$, $\lim_{t\to\infty} r_{ie} = \lim_{t\to\infty} v_{ie} = \lim_{t\to\infty} \omega_{ie} = [0\ 0\ 0]^T$.

The speed saturation function $sat(\cdot)$ is defined as follows:

$$sat(x) = \begin{cases} x & , xx/\|x\|_\infty < 1 \\ xx/\|x\|_\infty & , xx/\|x\|_\infty \geq 1 \end{cases} \tag{21}$$

The orbit controller of the six-degree-of-freedom satellite formation system is designed as follows:

$$\begin{aligned} f_i &= sat(-k_1 m_i \dot{r}_{ie} + k_1 m_i \omega_i^\times r_{ie} + m_i \omega_i^\times \dot{r}_{ie} - c_1(k_1 sat(r_{ie}) + \dot{r}_{ie})) \\ &\quad - \sum_{j=1}^{n} g_{ij}^1 l_1(k_1 r_{je}(t - T_{ij}) + \dot{r}_{je}(t - T_{ij})) - \sum_{j=1}^{n} g_{ij} d_1[(k_1 r_{ie} + \dot{r}_{ie}) - (k_1 r_{je}(t - T_{ij}) + \dot{r}_{je}(t - T_{ij}))] \end{aligned} \tag{22}$$

When $t \to \infty$, $\lim_{t\to\infty} r_{ie} = \lim_{t\to\infty} v_{ie} = [0\ 0\ 0]^T$ is satisfied in the controller where the control parameters $k_1$, $c_1$, $d_1$, $l_1$, $\gamma_1$ are constants greater than zero.

The attitude controller for the six-degree-of-freedom satellite formation system is designed as follows:

$$\begin{aligned} \tau_i &= sat(-\tfrac{1}{2}k_2 J_i(q_{ie}^\times + q_{ie0}I_3) + \omega_i^\times J_i \omega_i - J_i \omega_{ie}^\times T(Q_{ie})\omega_{id} \\ &\quad - c_2(k_2 sat(q_{ie}) + \omega_{ie})) - \sum_{j=1}^{n} g_{ij} l_2(k_2 q_{ie}(t - T_{ij}) + \omega_{ie}(t - T_{ij})) \\ &\quad - \sum_{j=1}^{n} g_{ij} d_2[(k_2 q_{ie} + \omega_{ie}) - (k_2 q_{ie}(t - T_{ij}) + \omega_{ie}(t - T_{ij}))] \end{aligned} \tag{23}$$

When $t \to \infty$, $\lim_{t\to\infty} \omega_{ie} = [0\ 0\ 0]^T$ and $\lim_{t\to\infty} Q_{ie} = [1\ 0\ 0\ 0]^T$ are satisfied in the controller, where the control parameters $k_2$, $c_2$, $d_2$, $l_2$, $\gamma_2$ are constants greater than zero.

### 3.4. Proof of Stability under Bounded Random Disturbance

In the above process, we designed a hierarchical saturated consensus cooperative controller of attitude orbit coupling. This section gives the proof of controller stability considering the influence of external interference. First, the following assumptions are given: the interference force $f_{di}$ and the interference moment $\tau_{di}$ are bounded and random and satisfy the following conditions, $\|f_{di}\| \leq \lambda_1$, $\|\tau_{di}\| \leq \lambda_2$; $\gamma_1$ and $\gamma_2$ are constants greater than zero; and $\|\bullet\|$ is the 2-norm of a matrix or vector.

Then, the position and attitude controller of bounded random disturbance is expressed as follows:

$$
\begin{aligned}
f_i &= sat(-k_1 m_i v_{ie} + k_1 m_i \omega_i^\times r_{ie} + m_i \omega_i^\times \dot{r}_{ie} - c_1(k_1 sat(r_{ie}) + \dot{r}_{ie}) \\
&\quad - \gamma_1 \mathrm{sgn}(k_1 sat(r_{ie}) + \dot{r}_{ie})) - \sum_{j=1}^{n} g_{ij}^1 l_1(k_1 r_{je}(t - T_{ij}) + \dot{r}_{je}(t - T_{ij})) \\
&\quad - \sum_{j=1}^{n} g_{ij} d_1 [(k_1 r_{ie} + \dot{r}_{ie}) - (k_1 r_{je}(t - T_{ij}) + \dot{r}_{je}(t - T_{ij}))]
\end{aligned}
\tag{24}
$$

$$
\begin{aligned}
\tau_i &= sat(-\tfrac{1}{2}k_2 J_i(q_{ie}^\times + q_{ie0}I_3) + \omega_i^\times J_i \omega_i - J_i \omega_{ie}^\times T(Q_{ie})\omega_{id} - c_2(k_2 sat(q_{ie}) + \omega_{ie}) \\
&\quad - \gamma_2 \mathrm{sgn}(k_2 sat(q_{ie}) + \omega_{ie})) - \sum_{j=1}^{n} g_{ij} l_2(k_2 q_{ie}(t - T_{ij}) + \omega_{ie}(t - T_{ij})) \\
&\quad - \sum_{j=1}^{n} g_{ij} d_2 [(k_2 q_{ie} + \omega_{ie}) - (k_2 q_{ie}(t - T_{ij}) + \omega_{ie}(t - T_{ij}))]
\end{aligned}
\tag{25}
$$

In the formula, $\gamma_1 \mathrm{sgn}(k_1 r_{ie} + \dot{r}_{ie}) \geq f_{di}$, $\gamma_2 \mathrm{sgn}(k_2 q_{ie} + \omega_{ie}) \geq \tau_{di}$, the symbolic function $\mathrm{sgn}(\bullet)$ is defined as follows:

$$
\mathrm{sgn}(x) = \begin{cases} 1 & , \ x > 0 \\ 0 & , \ x = 0 \\ -1 & , \ x < 0 \end{cases}
\tag{26}
$$

**Proof.** Relative to the orbit and attitude dynamics equations for the satellite formation system, the Lyapunov function $V$ is defined as follows:

$$
\begin{aligned}
V &= \tfrac{1}{2}\sum_{i=1}^{n} (k_1 r_{ie} + \dot{r}_{ie})^T m_i(k_1 r_{ie} + \dot{r}_{ie}) + \tfrac{1}{2}\sum_{i=1}^{n} r_{ie}^T \beta_1 r_{ie} \\
&\quad + \tfrac{1}{2}\sum_{i=1}^{n}\sum_{j=1}^{n} \int_{t-T_{ij}}^{t} (k_1 r_{je} + \dot{r}_{je})^T (k_1 r_{je} + \dot{r}_{je}) + \\
&\quad \tfrac{1}{2}\sum_{i=1}^{n} (k_2 q_{ie} + \omega_{ie})^T J_i(k_2 q_{ie} + \omega_{ie}) + \tfrac{1}{2}\sum_{i=1}^{n} q_{ie}^T \beta_2 q_{ie} \\
&\quad + \tfrac{1}{2}\sum_{i=1}^{n}\sum_{j=1}^{n} \int_{t-T_{ij}}^{t} (k_2 q_{ie} + \omega_{ie})^T (k_2 q_{ie} + \omega_{ie}) + \tfrac{1}{2}\sum_{i=1}^{n} (\beta_2 - \beta_2 q_{ie0})^2
\end{aligned}
\tag{27}
$$

In the Lyapunov function, $V$, $\beta_1$, and $\beta_2$ are constants greater than zero. To prove the stability of the hierarchical saturated uniform cooperative controller under bounded disturbance, the derivative of function $V$ is first obtained, and $f_i$ and $\tau_i$ are substituted into $\dot{V}$. The following expression is obtained:

$$
\begin{aligned}
\dot{V} &= -\sum_{i=1}^{n} (k_1 r_{ie} + \dot{r}_{ie})^T (\gamma_1 \mathrm{sgn}(k_1 r_{ie} + \dot{r}_{ie}) - f_{di}) \\
&\quad - c_1 k_1^2 \sum_{i=1}^{n} r_{ie}^T r_{ie} - c_1 \sum_{i=1}^{n} \dot{r}_{ie}^T \dot{r}_{ie} - (2c_1 k_1 - \beta)\sum_{i=1}^{n} r_{ie}^T \dot{r}_{ie} \\
&\quad - (g_{ij} d_1 - \tfrac{1}{2})\sum_{i=1}^{n}\sum_{j=1}^{n} (k_1 r_{je} + \dot{r}_{je})^T (k_1 r_{je} + \dot{r}_{je}) \\
&\quad - \tfrac{1}{2}\sum_{i=1}^{n}\sum_{j=1}^{n} (1 - \dot{T}_{ij})(k_1 r_{je}(t - T_{ij}) \\
&\quad + \dot{r}_{je}(t - T_{ij}))^T (k_1 r_{je}(t - T_{ij}) + \dot{r}_{je}(t - T_{ij})) \\
&\quad + \sum_{i=1}^{n} (k_1 r_{ie} + \dot{r}_{ie})^T (\sum_{j=1}^{n} g_{ij} d_1(1 - \tfrac{l_1}{d_1})[(k_1 r_{je}(t - T_{ij}) + \dot{r}_{je}(t - T_{ij}))]) \\
&\quad - \sum_{i=1}^{n} (k_2 q_{ie} + \omega_{ie})^T (\gamma_2 \mathrm{sgn}(k_2 q_{ie} + \omega_{ie}) - \tau_{di}) \\
&\quad - (g_{ij} d_2 - \tfrac{1}{2})\sum_{i=1}^{n}\sum_{j=1}^{n} (k_2 q_{ie} + \omega_{ie})^T (k_2 \dot{q}_{ie} + \dot{\omega}_{ie}) \\
&\quad - c_2 k_2^2 \sum_{i=i}^{n} q_{ie}^T q_{ie} - c_2 \sum_{i=i}^{n} \omega_{ie}^T \omega_{ie} - (2c_2 k_2 - \beta_2)\sum_{i=1}^{n} q_{ie}^T \omega_{ie} \\
&\quad - \tfrac{1}{2}\sum_{i=1}^{n}\sum_{j=1}^{n} (1 - \dot{T}_{ij})(k_2 q_{je}(t - T_{ij}) + \omega_{je}(t - T_{ij}))^T (k_2 q_{je}(t - T_{ij}) + \omega_{je}(t - T_{ij})) \\
&\quad + \sum_{i=1}^{n} (k_2 q_{ie} + \omega_{ie})^T \sum_{j=1}^{n} g_{ij} d_2(1 - \tfrac{l_2}{d_2})[(k_2 q_{ie}(t - T_{ij}) + \omega_{ie}(t - T_{ij}))]
\end{aligned}
\tag{28}
$$

When the designed hierarchical saturation consensus cooperative controller is stable, then the function $\dot{V} \leq 0$. Next, we analyse the function $\dot{V}$. Since the control parameters $k_1$, $k_2$, $c_1$, $c_2$, $d_1$, $d_2$, $l_1$, $l_2$ in the controller are all constants greater than 0, we have:

$$- c_1 k_1{}^2 \sum_{i=1}^{n} \boldsymbol{r}_{ie}{}^T \boldsymbol{r}_{ie} - c_1 \sum_{i=1}^{n} \dot{\boldsymbol{r}}_{ie}{}^T \dot{\boldsymbol{r}}_{ie} - c_2 k_2^2 \sum_{i=i}^{n} \boldsymbol{q}_{ie}{}^T \boldsymbol{q}_{ie} - c_2 \sum_{i=i}^{n} \boldsymbol{\omega}_{ie}{}^T \boldsymbol{\omega}_{ie} \leq 0 \tag{29}$$

When:

$$2c_1 k_1 - \beta = 0, 2c_2 k_2 - \beta_2 = 0,$$

Then

$$- (2c_1 k_1 - \beta) \sum_{i=1}^{n} \boldsymbol{r}_{ie}{}^T \dot{\boldsymbol{r}}_{ie} - (2c_2 k_2 - \beta_2) \sum_{i=1}^{n} \boldsymbol{q}_{ie}{}^T \boldsymbol{\omega}_{ie} = 0$$

Therefore, $\dot{V}$ satisfies the following inequality:

$$\begin{aligned}
\dot{V} < \ & -(g_{ij} d_1 - \tfrac{1}{2}) \sum_{i=1}^{n} \sum_{j=1}^{n} (k_1 \boldsymbol{r}_{je} + \dot{\boldsymbol{r}}_{je})^T (k_1 \boldsymbol{r}_{je} + \dot{\boldsymbol{r}}_{je}) \\
& - \tfrac{1}{2} \sum_{i=1}^{n} \sum_{j=1}^{n} (1 - \dot{T}_{ij})(k_1 \boldsymbol{r}_{je}(t - T_{ij}) + \dot{\boldsymbol{r}}_{je}(t - T_{ij}))^T (k_1 \boldsymbol{r}_{je}(t - T_{ij}) + \dot{\boldsymbol{r}}_{je}(t - T_{ij})) \\
& + \sum_{i=1}^{n} (k_1 \boldsymbol{r}_{ie} + \dot{\boldsymbol{r}}_{ie})^T \big( \sum_{j=1}^{n} g_{ij} d_1 ((1 - \tfrac{l_1}{d_1}))[(k_1 \boldsymbol{r}_{je}(t - T_{ij}) + \dot{\boldsymbol{r}}_{je}(t - T_{ij}))] \big) \\
& - (g_{ij} d_2 - \tfrac{1}{2}) \sum_{i=1}^{n} \sum_{j=1}^{n} (k_2 \boldsymbol{q}_{ie} + \boldsymbol{\omega}_{ie})^T (k_2 \dot{\boldsymbol{q}}_{ie} + \dot{\boldsymbol{\omega}}_{ie}) \\
& - \tfrac{1}{2} \sum_{i=1}^{n} \sum_{j=1}^{n} (1 - \dot{T}_{ij})(k_2 \boldsymbol{q}_{je}(t - T_{ij}) + \boldsymbol{\omega}_{je}(t - T_{ij}))^T (k_2 \boldsymbol{q}_{je}(t - T_{ij}) + \boldsymbol{\omega}_{je}(t - T_{ij})) \\
& + \sum_{i=1}^{n} (k_2 \boldsymbol{q}_{ie} + \boldsymbol{\omega}_{ie})^T \sum_{j=1}^{n} g_{ij} d_2 (1 - \tfrac{l_2}{d_2})[(k_2 \boldsymbol{q}_{ie}(t - T_{ij}) + \boldsymbol{\omega}_{ie}(t - T_{ij}))]
\end{aligned} \tag{30}$$

According to the formula $a^2 + b^2 \leq 2ab$, then

$$\begin{aligned}
\sum_{i=1}^{n} \sum_{j=1}^{n} [(k_1 \boldsymbol{r}_{ie} & + \dot{\boldsymbol{r}}_{ie})^T (k_1 \boldsymbol{r}_{je}(t - T_{ij}) + \dot{\boldsymbol{r}}_{je}(t - T_{ij}) + (k_2 \boldsymbol{q}_{ie} + \boldsymbol{\omega}_{ie})^T (k_2 \boldsymbol{q}_{ie}(t - T_{ij}) + \boldsymbol{\omega}_{ie}(t - T_{ij})))] \\
& \leq \tfrac{1}{2} \big( \sum_{i=1}^{n} \sum_{j=1}^{n} (k_1 \boldsymbol{r}_{ie} + \dot{\boldsymbol{r}}_{ie})^T (k_1 \boldsymbol{r}_{ie} + \dot{\boldsymbol{r}}_{ie}) + + \sum_{j=1}^{n} \sum_{i=1}^{n} (k_2 \boldsymbol{q}_{ie} + \boldsymbol{\omega}_{ie})^T (k_2 \boldsymbol{q}_{ie} + \boldsymbol{\omega}_{ie}) \\
& + \sum_{i=1}^{n} \sum_{j=1}^{n} ((k_1 \boldsymbol{r}_{je}(t - T_{ij}) + \dot{\boldsymbol{r}}_{je}(t - T_{ij}))^T (k_1 \boldsymbol{r}_{je}(t - T_{ij}) + \dot{\boldsymbol{r}}_{je}(t - T_{ij}) \\
& + \sum_{j=1}^{n} \sum_{i=1}^{n} (k_2 \boldsymbol{q}_{ie}(t - T_{ij}) + \boldsymbol{\omega}_{ie}(t - T_{ij}))^T (k_2 \boldsymbol{q}_{ie}(t - T_{ij}) + \boldsymbol{\omega}_{ie}(t - T_{ij})))
\end{aligned} \tag{31}$$

Then, the following inequality $\dot{V}$ can be satisfied:

$$\begin{aligned}
\dot{V} \leq \ & -\tfrac{1}{2} \big[ (g_{ij} d_1 - 1) \sum_{i=1}^{n} \sum_{j=1}^{n} (k_1 \boldsymbol{r}_{je} + \dot{\boldsymbol{r}}_{je})^T (k_1 \boldsymbol{r}_{je} + \dot{\boldsymbol{r}}_{je}) \\
& + (1 - \dot{T}_{ij} - g_{ij} d_1 (1 - \tfrac{l_1}{d_1})) \sum_{i=1}^{n} \sum_{j=1}^{n} (k_1 \boldsymbol{r}_{je}(t - T_{ij}) + \dot{\boldsymbol{r}}_{je}(t - T_{ij}))^T (k_1 \boldsymbol{r}_{je}(t - T_{ij}) + \dot{\boldsymbol{r}}_{je}(t - T_{ij})) \\
& + (g_{ij} d_2 - 1) \sum_{i=1}^{n} \sum_{j=1}^{n} (k_2 \boldsymbol{q}_{ie} + \boldsymbol{\omega}_{ie})^T (k_2 \dot{\boldsymbol{q}}_{ie} + \dot{\boldsymbol{\omega}}_{ie}) \\
& + (1 - \dot{T}_{ij} - g_{ij} d_2 (1 - \tfrac{l_2}{d_2})) \sum_{i=1}^{n} \sum_{j=1}^{n} (k_2 \boldsymbol{q}_{je}(t - T_{ij}) + \boldsymbol{\omega}_{je}(t - T_{ij}))^T (k_2 \boldsymbol{q}_{je}(t - T_{ij}) + \boldsymbol{\omega}_{je}(t - T_{ij}))
\end{aligned} \tag{32}$$

For $\dot{T}_{ij}$, when the extension time is a fixed length, then $\dot{T}_{ij} = 0$; when the extension time is time-varying, parameters $1 - \dot{T}_{ij} - g_{ij} d_1 (1 - \tfrac{l_1}{d_1})$ and $1 - \dot{T}_{ij} - g_{ij} d_2 (1 - \tfrac{l_2}{d_2})$ can be

set jointly; when $g_{ij}d_1 - 1 \geq 0$, $g_{ij}d_2 - 1 \geq 0$, $1 - \dot{T}_{ij} - g_{ij}d_1(1 - \frac{l_1}{d_1}) \geq 0$, $1 - \dot{T}_{ij} - g_{ij}d_2(1 - \frac{l_2}{d_2}) \geq 0$, then $\dot{V} \leq 0$. The results show that the designed hierarchical saturation consensus cooperative controller is stable under bounded disturbance. □

## 4. Simulation Analysis

During the Earth observation mission of an SAR satellite formation, it is assumed that four SAR satellites form a spatially symmetric elliptical formation centred on the reference star. The position and attitude of the reference star are ideal. There is a certain initial deviation in the formation configuration. The satellite formation manoeuvre control is required to ensure that the satellite maintains the four-star space symmetric elliptical configuration as much as possible. The attitude of the satellite formation should be consistent with the reference star as much as possible. The attitude angle of the formation satellite and reference satellite are kept within the error range of 0.01°. The attitude angular velocity of the formation satellite and reference satellite are kept within the error range of 0.01°/s. In the control process, there will be communication delays and changes in topology between the satellites in the formation. To solve the above problems, a hierarchical saturated consistency cooperative controller is designed in this paper. To verify the effectiveness of the controller, the following simulation verification is carried out. Considering the on-orbit operation of SAR satellite formation and other factors, simulation parameter settings are given in Table 1.

**Table 1.** SAR formation simulation parameters.

| Simulation Parameters | |
|---|---|
| orbital elements of the reference star | $[6988.01km\ 0.00091\ 97.11°\ 180°\ 90°\ 0°]$ |
| quaternion initial value of the reference star | $[0.5\ 0.5\ 0.5\ 0.5]$ |
| angular velocity of the reference star (°/s) | $[0.1\cos(0.1\pi t)\ 0.1\cos(0.1\pi t)\ 0.1\cos(0.1\pi t)]$ |
| formation configuration parameters | $[522.2836km\ 200m\ 90*i°\ 0°\ 581.9080m]$ |
| moment of inertia of satellites (kg m²) | $J_1 = J_2 = J_3 = J_4 = \begin{bmatrix} 2.20\ 0.12\ 0.15 \\ 0.12\ 2.20\ 0.40 \\ 0.15\ 0.40\ 3.01 \end{bmatrix}$ |
| weight of satellites (kg) | $m_1 = m_2 = m_3 = m_4 = 100$ |
| interference force of satellites (N) | $f_{d1} = f_{d2} = f_{d3} = f_{d4} = 0.01[\sin(0.1\pi t)\ \sin(0.1\pi t)\ \cos(0.1\pi t)]$ |
| interference torque of satellites (N m) | $\tau_{d1} = \tau_{d2} = \tau_{d3} = \tau_{d4} =$ $0.001[\sin(0.1\pi t)\ \sin(0.1\pi t)\ \cos(0.1\pi t)]$ |
| initial relative position of satellites (m) | $[-521.5\ -0.61\ 523.5\ -0.6;\ 582.6\ 1627.9\ 583\ -463.7;$ $0.23\ 199.3\ 0.22\ -199.6]$ |
| initial velocity of satellites (m/s) | $[-0.015\ 0.589\ -0.0013\ -0.58;\ 1.17\ -0.011\ -1.172\ -0.014;$ $0.119\ 0.016\ -0.115\ 0.014]$ |
| initial quaternion of satellites | $\frac{1}{\sqrt{10}}[1\ -2\ -2\ -1;1\ -1\ 2\ -2;2\ -1\ 1\ -2;2\ -2\ 1\ -1]$ |
| initial angular velocity of satellites (°/s) | $[-4.01\ 1.72\ 1.15;\ 2.86\ 3.44\ -4.01;$ $-1.72\ -1.72\ 2.86;\ 1.15\ 2.30\ -1.72]$ |

Considering the constraints for the SAR satellite formation actuators and measurement mechanisms, the simulation sets the following parameters: each output force amplitude of the satellite orbit control engine is 0.5 N; each output torque amplitude of the attitude actuator is 0.2 N m; the maximum velocity error of the satellite is 0.2 m/s; the maximum

angular velocity of the attitude manoeuvre is 2°/s; the position determination error is 0.001 m; the velocity determination error is 0.001 m/s; the attitude angle determination error is 0.005 deg; the determination error for the attitude angular velocity is 0.001°/s; the error in the measurement mechanism is 0.001; the error in the attitude sensor is 0.001.

In order to analyse the performance of the configuration maintenance and relative attitude maintenance, the following indicators are introduced:

$$\rho_1 = \|\boldsymbol{r}_{1e} - \boldsymbol{r}_{2e}\| + \|\boldsymbol{r}_{1e} - \boldsymbol{r}_{3e}\| + \|\boldsymbol{r}_{1e} - \boldsymbol{r}_{4e}\| + \|\boldsymbol{r}_{2e} - \boldsymbol{r}_{3e}\| + \|\boldsymbol{r}_{2e} - \boldsymbol{r}_{4e}\| + \|\boldsymbol{r}_{3e} - \boldsymbol{r}_{4e}\| \tag{33}$$

$$\rho_2 = \|\boldsymbol{E}_{1e} - \boldsymbol{E}_{2e}\| + \|\boldsymbol{E}_{1e} - \boldsymbol{E}_{3e}\| + \|\boldsymbol{E}_{1e} - \boldsymbol{E}_{4e}\| + \|\boldsymbol{E}_{2e} - \boldsymbol{E}_{3e}\| + \|\boldsymbol{E}_{2e} - \boldsymbol{E}_{4e}\| + \|\boldsymbol{E}_{3e} - \boldsymbol{E}_{4e}\| \tag{34}$$

where $\rho_1$ is the 2-norm sum of the difference between the current position of the slave star and the reference star, and $\rho_2$ is the 2-norm sum of the difference between the current attitude angle of the slave star and the reference star. When the designed controller is stable, the values of $\rho_1$ and $\rho_2$ will become smaller and smaller; when the final $\rho_1$ and $\rho_2$ are smaller, it means that the controller has higher performance of configuration maintenance and relative attitude maintenance.

In this paper, the simulation results of attitude orbit coupling control without consistent cooperative control are analysed and compared. The simulation conditions are the same except for the consistent cooperative control items.

Case 1: Verify the performance of the hierarchical saturated consistency controller without considering the consistency coordination term.

The controllers without consistent collaborative control are:

$$\boldsymbol{f}_i = sat(-k_1 m_i \boldsymbol{v}_{ie} + k_1 m_i \boldsymbol{\omega}_i^\times \boldsymbol{r}^{ie} + m_i \boldsymbol{\omega}_i^\times \dot{\boldsymbol{r}}_{ie} - c_1(k_1 sat(\boldsymbol{r}_{ie}) + \dot{\boldsymbol{r}}_{ie}) - \gamma_1 \text{sgn}(k_1 \boldsymbol{r}_{ie} + \dot{\boldsymbol{r}}_{ie})) \tag{35}$$

$$\begin{aligned}\boldsymbol{\tau}_i &= sat(-\tfrac{1}{2}k_2 \boldsymbol{J}_i(\boldsymbol{q}_{ie}^\times + q_{ie0}\boldsymbol{I}_3) + \boldsymbol{\omega}_i^\times \boldsymbol{J}_i \boldsymbol{\omega}_i - \boldsymbol{J}_i \boldsymbol{\omega}_{ie}^\times T(\boldsymbol{Q}_{ie})\boldsymbol{\omega}_{id} \\ &\quad -c_2(k_2 sat(\boldsymbol{q}_{ie}) + \boldsymbol{\omega}_{ie}) - \gamma_2 \text{sgn}(k_2 \boldsymbol{q}_{ie} + \boldsymbol{\omega}_{ie})\,)\end{aligned} \tag{36}$$

The parameters in the controller are selected as follows: $k_1 = k_2 = 1$, $c_1 = c_2 = 1$, $\gamma_1 = 0.28$, $\gamma_2 = 0.014$.

The simulation results for non-consistent collaborative control are as follows.

As shown in Figures 3 and 4, the relative position errors and velocity errors for SAR satellite formation can reach the steady state within 200 s under the action of track controllers due to the limited output of the satellite orbit control engine. Due to the presence of spatial uncertainty and continuous interference force, the position control accuracy of the controller without the cooperative item is 0.003 m, and the speed control accuracy of the controller without the cooperative item is 0.005 m/s.

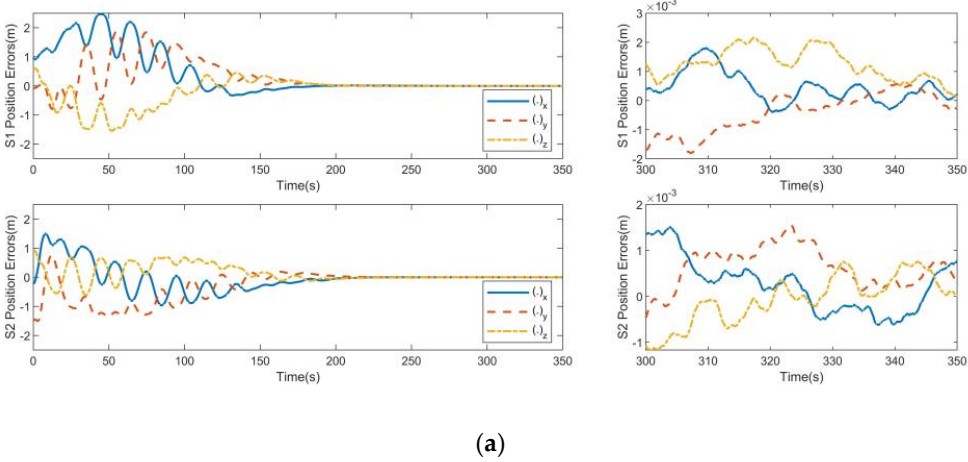

(a)

**Figure 3.** *Cont.*

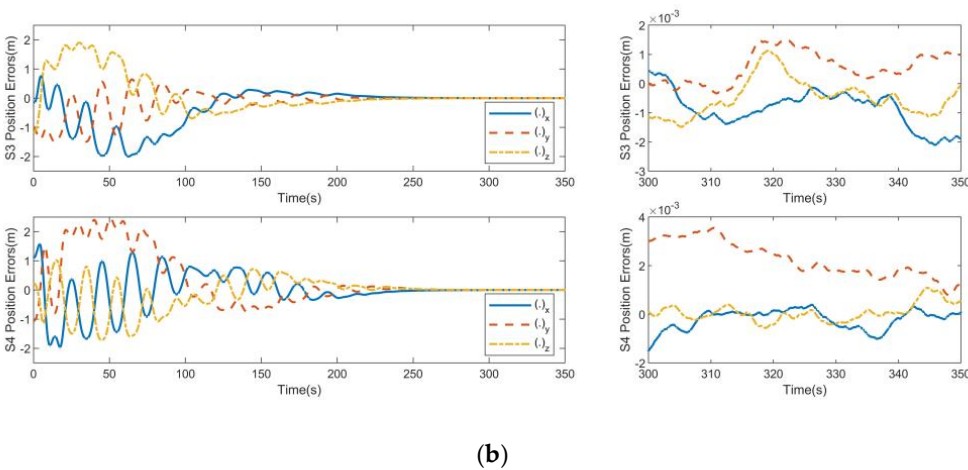

(**b**)

**Figure 3.** Position errors for non-consistent collaborative control: (**a**) S1 position errors and S2 position errors; (**b**) S3 position errors and S4 position errors.

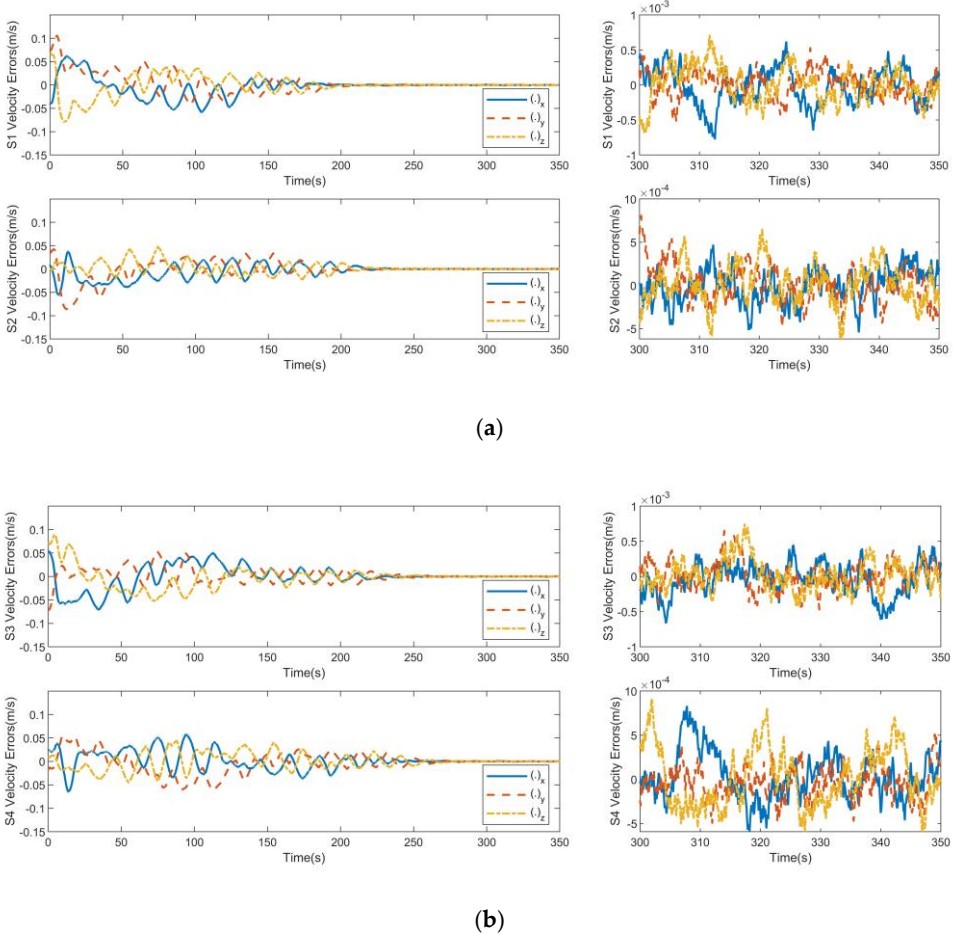

**Figure 4.** Velocity errors for non-consistent collaborative control: (**a**) S1 velocity errors and S2 velocity errors; (**b**) S3 velocity errors and S4 velocity errors.

As shown in Figures 5 and 6, the attitude angle errors and attitude angular velocity errors can reach the steady state within 25 s under the action of attitude controllers. Due to the presence of spatial uncertainty and continuous interference torque, the attitude angle control accuracy of the controller without the cooperative item is $0.01°$, and the attitude angular velocity control accuracy of the controller without the cooperative item is $0.005°/s$.

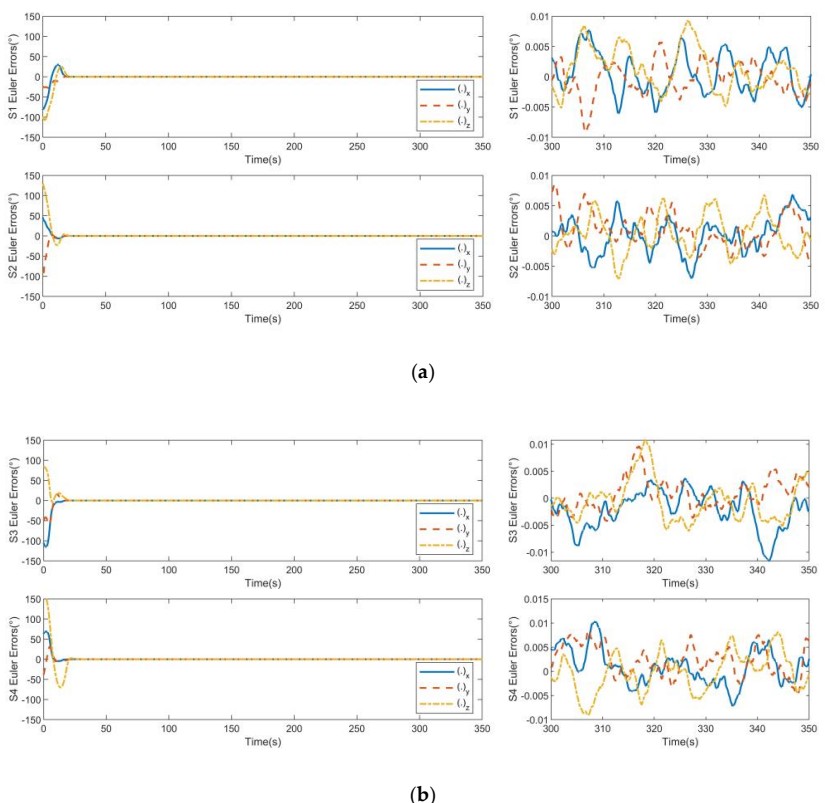

**Figure 5.** Euler errors for non-consistent collaborative control: (**a**) S1 Euler errors and S2 Euler errors; (**b**) S3 Euler errors and S4 Euler errors.

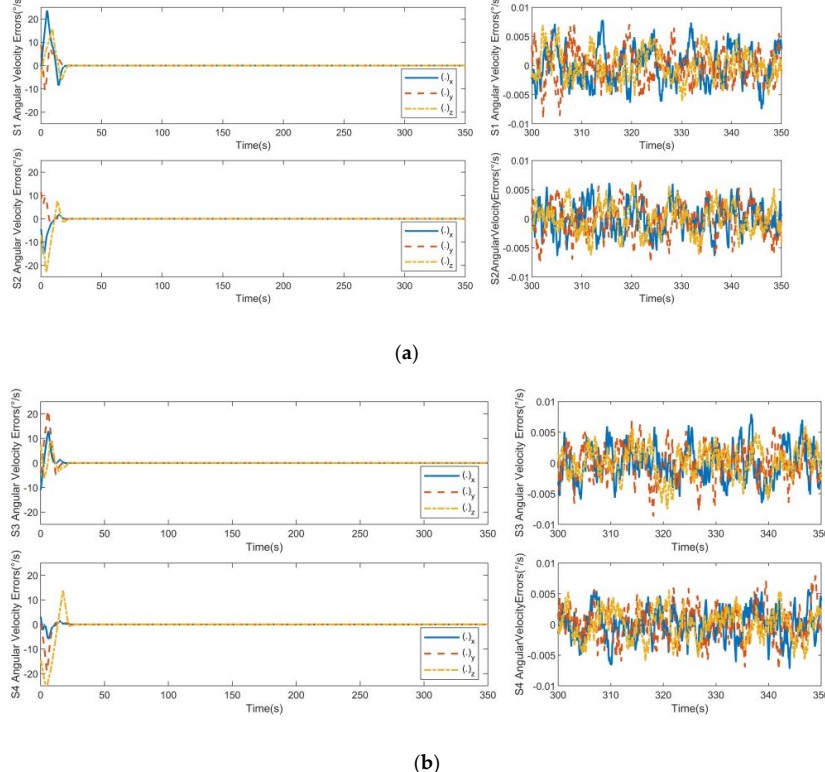

**Figure 6.** Angular velocity errors for non-consistent collaborative control: (**a**) S1 angular velocity errors and S2 angular velocity errors; (**b**) S3 angular velocity errors and S4 angular velocity errors.

As shown in Figures 7 and 8, the control force and control torque are relatively large due to the large initial error at the initial stage of control. The orbit control can reach the steady state within 250 s. The attitude control can reach the steady state within 30 s. After reaching the stable state, the controller continues to output to offset the external disturbance and maintain the stable state of the system. The magnitude of the control output is the same as the disturbance, which is in line with the actual situation.

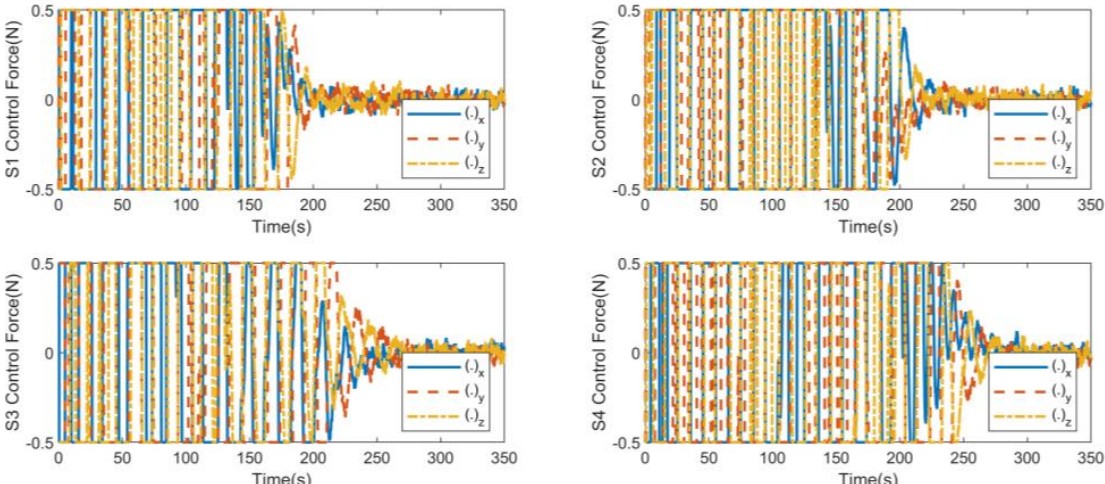

**Figure 7.** S1, S2, S3, S4 control force for non-consistent collaborative control.

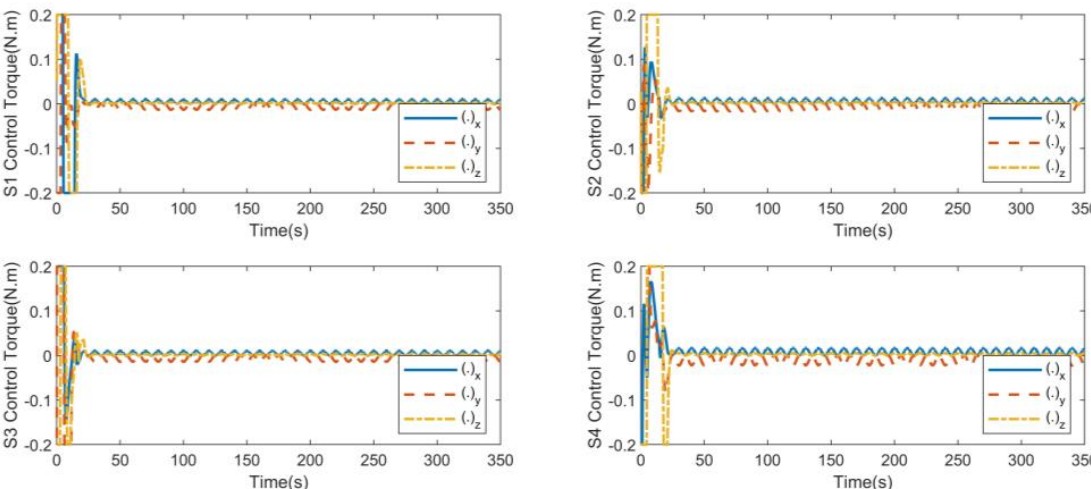

**Figure 8.** S1, S2, S3, S4 control torque for non-consistent collaborative control.

Case 2: Verify the performance of hierarchical saturated consistent collaborative controller under time-varying delay and switching topology.

Assuming that the communication network is stable and the delay matrix between satellites is a variable value, the delay matrix is set as:

$$
T_{ij} = \begin{bmatrix}
0 & 1 + 0.6\sin(0.1\pi t) & 1 - 0.5\cos(0.1\pi t) & 1 + 0.5\cos(0.1\pi t) \\
1 + 0.9\cos(0.1\pi t) & 0 & 1 + 0.5\sin(0.1\pi t) & 1 + 0.7\cos(0.1\pi t) \\
1 + 0.6\sin(0.1\pi t) & 1 + 0.2\cos(0.1\pi t) & 0 & 1 + 0.5\sin(0.1\pi t) \\
1 + 0.5\cos(0.1\pi t) & 1 - 0.5\sin(0.1\pi t) & 1 - 0.5\cos(0.1\pi t) & 0
\end{bmatrix} s
$$

In actual engineering, the delay of the satellite is smaller than that set. The large delay is set to verify the performance of the designed controller. Assuming that the communication topology changes, the specific changes are as follows:

When $t = [0, 100]$, $i \neq j$, then
$G_{ij} = [1\,1\,1\,1\,;\,1\,1\,1\,1\,;\,1\,1\,1\,1\,;\,1\,1\,1\,1]$
When $t = (100, 200]$, $i \neq j$, then
$G_{ij} = [0\,1\,0\,1\,;\,1\,0\,1\,0\,;\,0\,1\,0\,1\,;\,1\,0\,1\,0]$.
When $t = [200, 250]$, $i \neq j$, then
$G_{ij} = [0\,1\,0\,0\,;\,0\,0\,1\,0\,;\,0\,0\,0\,1\,;\,1\,0\,0\,0]$.

The parameters in the controller are selected as follows: $k_1 = k_2 = 1$, $c_1 = c_2 = 1$, $\gamma_1 = 0.28$, $\gamma_2 = 0.014$, $d_1 = d_2 = 2$, $l_1 = l_2 = 3$.

The simulation results from consistent collaborative control are as follows.

As shown in Figures 9 and 10, the relative position errors and velocity errors for SAR satellite formation can reach the steady state within 180 s under the action of the track controller due to the limited output of the satellite orbit control engines. Due to the presence of spatial uncertainty and continuous interference force, the position control accuracy of the controller of the cooperative item is 0.003 m and the speed control accuracy of the controller of the cooperative item is 0.001 m/s.

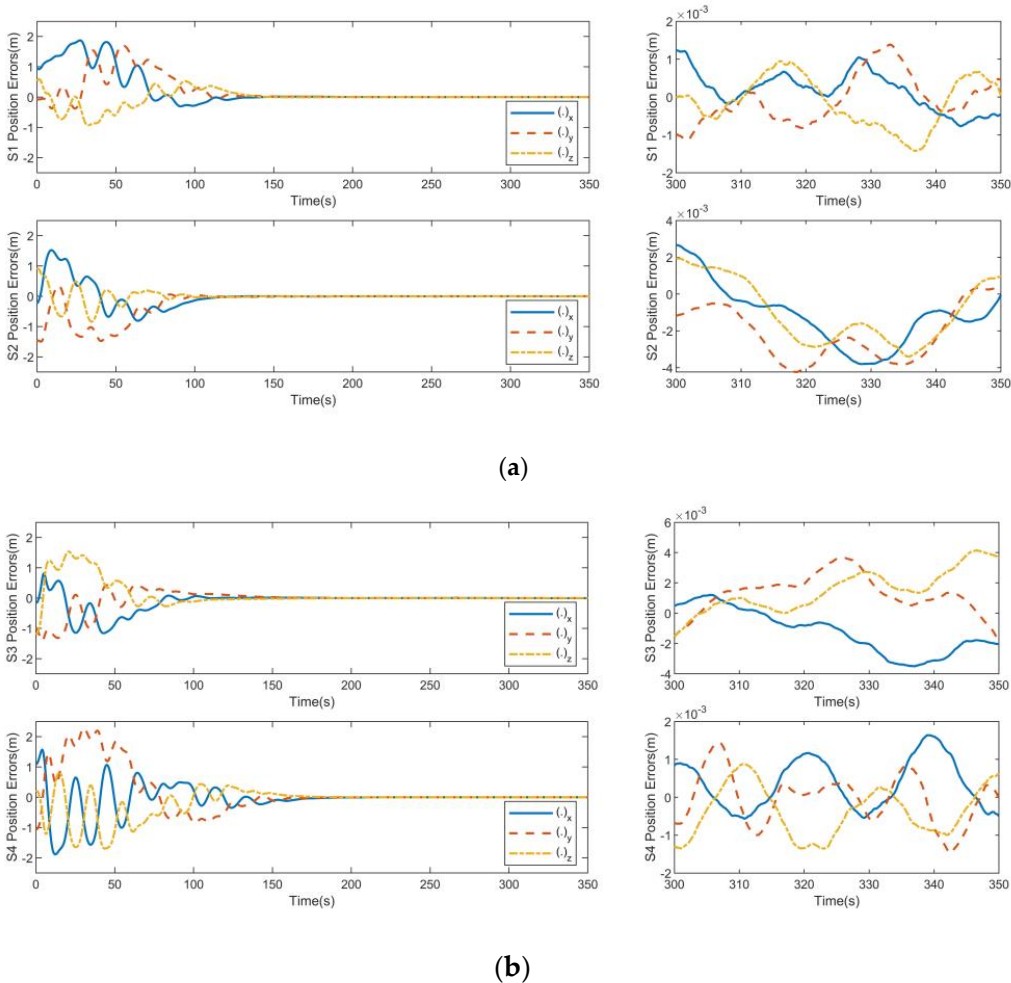

(**a**)

(**b**)

**Figure 9.** Position errors for consistent collaborative control: (**a**) S1 position errors and S2 position errors; (**b**) S3 position errors and S4 position errors.

As shown in Figures 11 and 12, the attitude angle error and attitude angular velocity error can reach the steady state within 23 s under the action of attitude controllers. Due to the presence of spatial uncertainty and continuous interference torque, the attitude angle control accuracy of the controller of the cooperative item is 0.002°, and the attitude angular velocity control accuracy of the controller of the cooperative item is 0.001°/s.

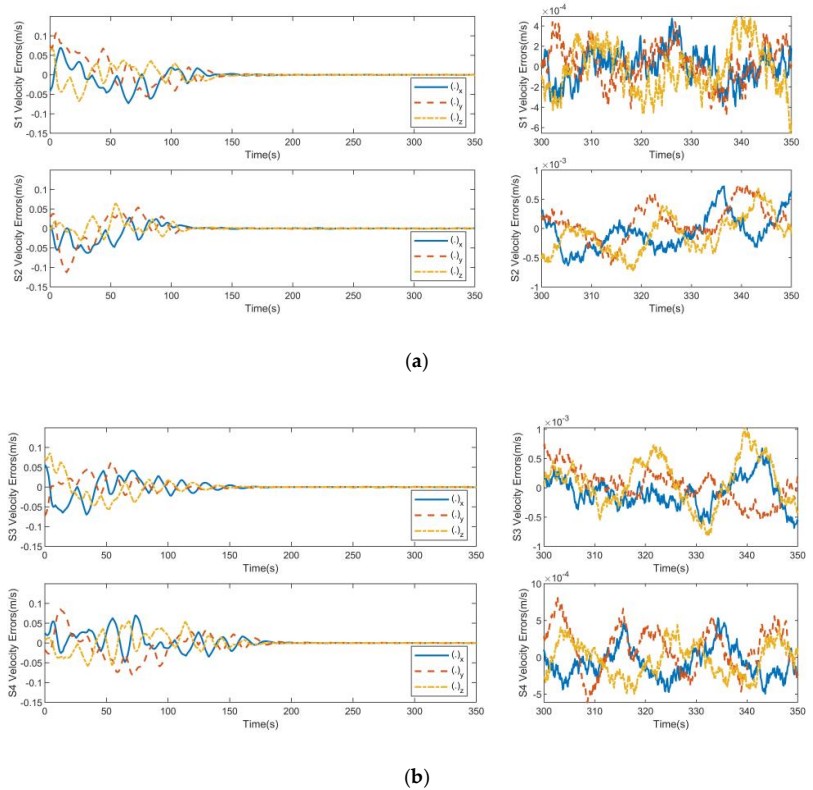

(**a**)

(**b**)

**Figure 10.** Velocity errors for consistent collaborative control. (**a**) S1 velocity errors and S2 velocity errors; (**b**) S3 velocity errors and S4 velocity errors.

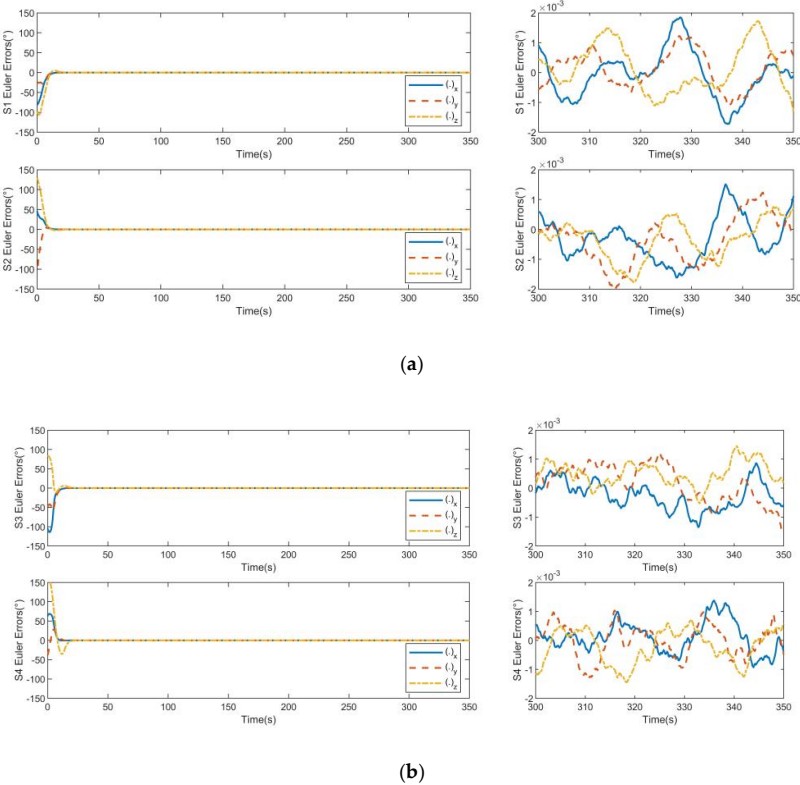

(**a**)

(**b**)

**Figure 11.** Euler errors for consistent collaborative control: (**a**) S1 Euler errors and S2 Euler errors; (**b**) S3 Euler errors and S4 Euler errors.

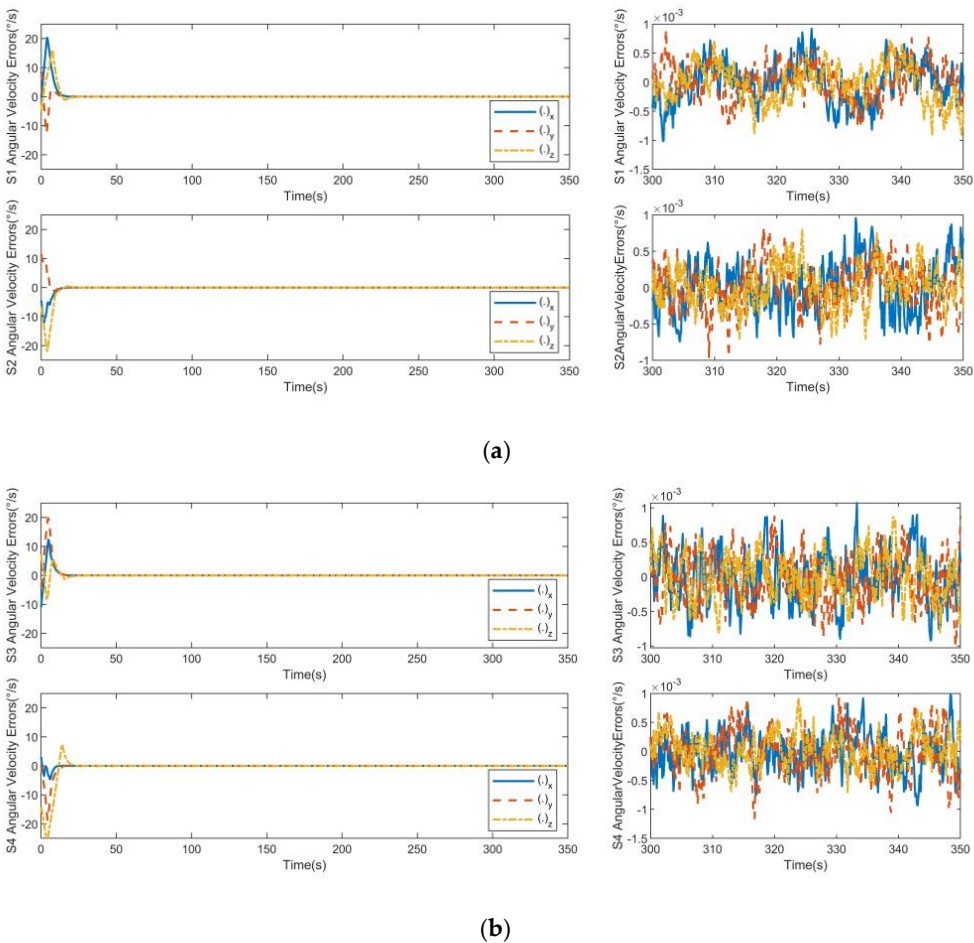

**Figure 12.** Angular velocity errors for consistent collaborative control: (**a**) S1 angular velocity errors and S2 angular velocity errors; (**b**) S3 angular velocity errors and S4 angular velocity errors.

As shown in Figures 13 and 14, the control force and control torque are relatively large due to the large initial error at the initial stage of control. The orbit control reaches the steady state within 180 s. The attitude control reaches the steady state within 23 s. After reaching the stable state, the controller continues to output to offset the external disturbance and maintain the stable state of the system. The magnitude of the control output is the same as that of the disturbance, which is in line with the actual situation.

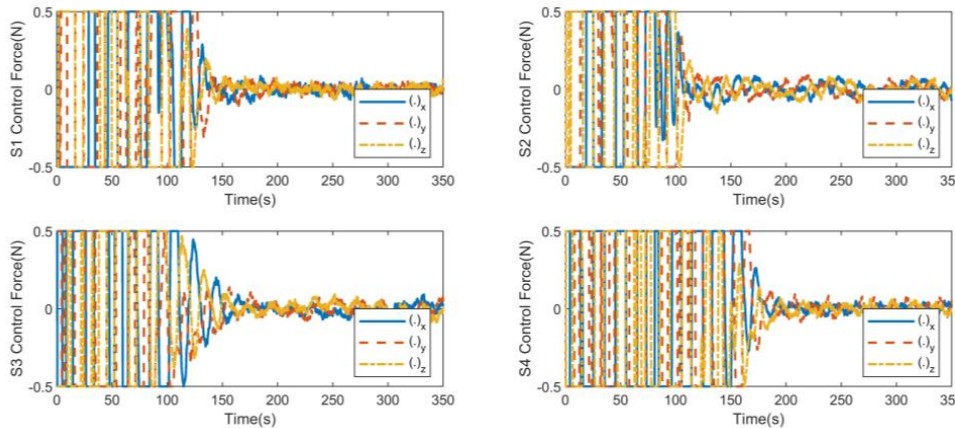

**Figure 13.** S1, S2, S3, S4 control force for consistent collaborative control.

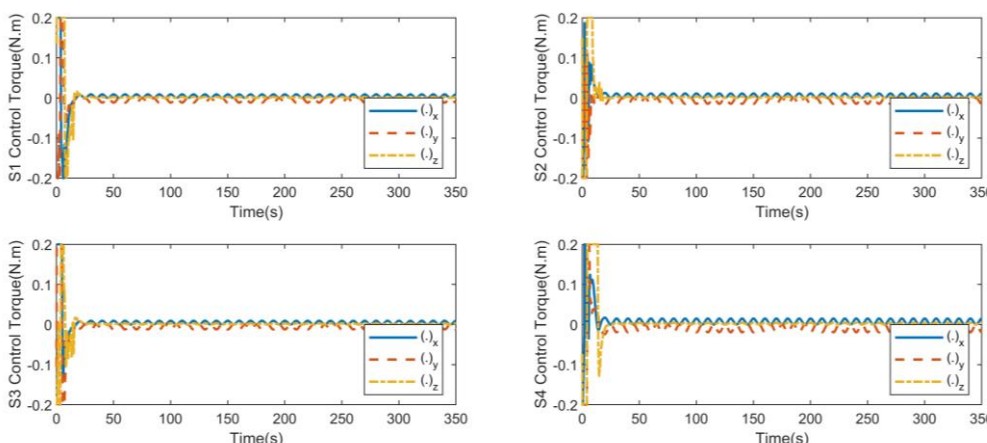

**Figure 14.** S1, S2, S3, S4 control torque for consistent collaborative control.

Figure 15 shows the performance curve under the separate tracking control of the relative position. Figure 16 shows the performance curve under the separate tracking control of the attitude coordination items. From the comparison curves in the figure, it can be seen that the situation with cooperative terms can better maintain the state consistency while realizing tracking. At the same time, with and without cooperative terms in the steady-state stage, it can be seen that the performance difference between configuration maintenance and relative attitude maintenance is obvious when all satellites converge to the steady-state value.

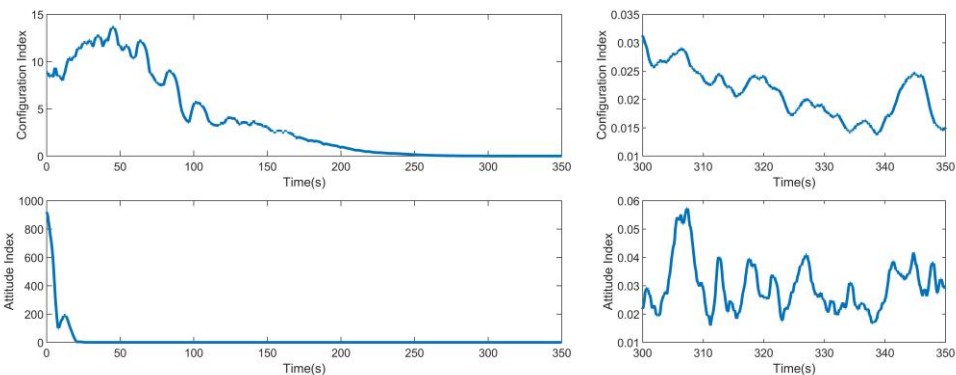

**Figure 15.** Configuration maintenance index and relative attitude index of non-consistent collaborative control for SAR satellite formation.

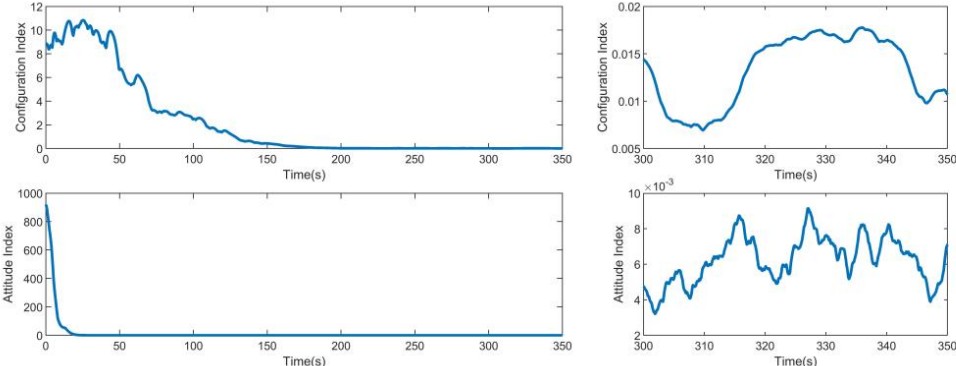

**Figure 16.** Configuration maintenance index and relative attitude index of consistent collaborative control for SAR satellite formation.

The performances of the above two controllers are shown in Table 2. Comparing various parameters between non-consistent collaborative control and consistency collaborative control, the following conclusions can be drawn: in the presence of communication constraints, actuator capacity constraints, and external uncertain interference, the case 2 controller can achieve high orbit and attitude control accuracy. The simulation verifies the effectiveness of the consistent cooperative control method of attitude orbit coupling to SAR satellite formations.

**Table 2.** Performance comparison of coordinated control with or without consistency.

| Comparison Item | Non-Consistent Collaborative Control | Consistency Collaborative Control |
|---|---|---|
| Position control accuracy | 0.003 m | 0.003 m |
| Velocity control accuracy | 0.005 m/s | 0.001 m/s |
| Attitude angle control accuracy | 0.01° | 0.002° |
| Angular velocity control accuracy | 0.005°/s | 0.001°/s |
| Orbit control time | Stable to 200 s | Stable to 180 s |
| Attitude control time | Stable to 25 s | Stable to 23 s |
| Configuration maintenance index | Convergence to 0.02 | Convergence to 0.015 |
| Relative attitude index | Convergence to 0.03 | Convergence to 0.006 |

## 5. Conclusions

Firstly, this paper establishes the attitude orbited coupling dynamics model of SAR satellite formation to express the influence of attitude to orbit by establishing the relative orbit dynamics equation in the satellite body coordinated system. Secondly, this paper designs a hierarchical saturated consensus cooperative controller of communication delays, topology switching, actuator capacity constraints, and external uncertain disturbances. Finally, it proves the stability of the controller by the Lyapunov direct method. The simulation results show that the hierarchical saturation consistency cooperative controller based on the attitude orbit coupling control model designed in this paper can meet the requirements of configuration maintenance accuracy in the ground target detection task when SAR satellite formation control has relatively large initial errors.

Two open points remain for the future research. The first one is the change of mass characteristics for the satellite. This paper did not consider the change of mass characteristics, to focus on analysing only the short-term constant behaviour. However, by checking a posteriori, in some cases the change of mass characteristics did have some negative influence on the consistency cooperative control. The second problem to address is the attitude and orbit control of formations with long baselines. The methodology proposed in this paper, i.e., first designing controllers and then solving delays and structural changes of the communication, can be used to address a more general problem.

**Author Contributions:** B.H. provided the overall research ideas for the thesis and put forward constructive suggestions for the writing of the thesis; S.Z. participated in the discussion of the research ideas of the thesis and completed the simulation experiments and manuscript writing; Y.W. and Z.C. provided instructions for the research and thesis writing. All authors have read and agreed to the published version of the manuscript.

**Funding:** This research was funded by National Natural Science Foundation of China (Nos. 61973153 and 6207316).

**Data Availability Statement:** The data presented in this study are available on request from the corresponding author.

**Conflicts of Interest:** The authors declare no conflict of interest.

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
