# Peer review of "Research into a Consistency Cooperative Control Method for Attitude Orbit Coupling of SAR Satellite Formations under Communication Constraints"

_aerospace, doi:10.3390/aerospace9100556_

Round 1

Reviewer 1 Report

The paper addresses the design of a control method for attitude orbit coupling satellite formation under several constraints, such as delays in the intersatellite communication and topology switching, considering also actuator limitations and external disturbances. Additionally, proof of the stability of the designed controller is given. The performance of two controllers with and without consistent cooperative control items are verified and compared by simulation analysis. 

general comments

The work is consistent at a conceptual and mathematical level, with all the steps required for designing the controller correct and well detailed. However, a clear definition of the scenario and conditions, for which the problems highlighted in the work become relevant, is not given. For example, what are the separations between the member satellites in the formation and with the reference satellite, resulting in a non-negligible communication delay? But also, what is the required accuracy in the consistency between the attitude of reference satellite and member satellites? As well, what are the maximum allowable errors in actuator and sensing mechanisms for which the effects on formation control can be ignored? It is suggested to enhance the clarity of the work on the above points. 

It is suggested to improve the general readability of the text. The quality of images should be improved too, in order to have a clearer view of the results. 

SPECIFIC COMMENTS

1. Page 3, line 81

There is a repetition of the same sentence in line 80

2. Figure 1, line 87 

Please make the symbols more readable. 

3. Page 3, line 90

Please specify that the equation is referred to the i-th member satellite 

4. Page 4, line 105

It is suggested to change "formula" in "equation"

5. Page 4, line 105

Please check that equation 4 is obtained by substituting equation 1 into equation 2. Should not equation 2 be substituted into equation 3?

6. Page 4, line 115 

It is suggested to add a sentence about the consequences of not assuming the reference satellite in an ideal state. 

7. Page 5, line 135 

Please rewrite for clarity

8. Page 5, line 136 

Please specify the terms with the suffix "ie". In the text, only the relative terms with the suffix "e" have been described.

9. Page 11, line 251 

It is recommended to specify what is the ideal attitude for the reference satellite (e.g., nadir pointing).

10. Page 11, lines 254 and 255

Please consider specifying how much the attitude of satellite formation should be consistent with the reference satellite. 

11. Page 16, line 342 

It is suggested to clarify how the delay matrix has been derived

Author Response

Dear Reviewer,,

We are very grateful to the reviewers for their comments, which are extremely valuable to improve the quality of our manuscript. We have considered seriously the suggestions and discussed how to revise the manuscript. We have carefully checked that the references cited in the revised manuscript are related to the content of the manuscript. The revisions are presented in detail in the following. In order to facilitate you to quickly find the modified parts of the manuscript, we use the "track changes" function in MS word. In addition, we also highlight the modified main parts in the revised manuscript. In addition, as the reviewer pointed out that there are many language problems in the manuscript I submitted, the revised manuscript we submitted this time has been provided with professional language editing services, and we will attach relevant certificates at the end.

  1. Page 3, line 81

There is a repetition of the same sentence in line 80

Answer: The author has deleted the duplicate sentences in the revised manuscript.(line 92)

  1. Figure 1, line 87

Please make the symbols more readable.

Answer: The author has supplemented the description of symbols in the revised manuscript, and the description of symbols is as follows: (line 86-100)

 is the th satellite body coordinate system, in which the three-axis direction is consistent with the satellite inertia axis.  is the centroid for the th satellite of the formation,  is the position vector from the reference satellite centroid to the Earth center,  is the position vector from the centroid for the th satellite of the formation to the reference satellite, and  is the position vector from the centroid for the th satellite of the formation to the Earth center. In the full paper, the ‘’ subscript at the lower right of the satellite state parameters is uniformly expressed as the state parameters of the i th satellite.

  1. Page 3, line 90

Please specify that the equation is referred to the i-th member satellite

Answer: The author has added a note to the i-th member satellite in the revised manuscript, and the added note is as follows: (line 98)

The dynamic and kinematic equations of i-th member satellite represents the any satellite in the formation.

  1. Page 4, line 105

It is suggested to change "formula" in "equation"

Answer: The author has corrected this error in the revised manuscript, and the corrected sentence is as follows: (line 120)

By substituting equation 1 and equation 2 into equation 3, the orbit dynamics equation containing the satellite rotation angular velocity is obtained as follows:

  1. Page 4, line 105

Please check that equation 4 is obtained by substituting equation 1 into equation 2. Should not equation 2 be substituted into equation 3?

Answer: Equation 2 should be substituted into equation 3. The author has corrected this error in the revised manuscript, and the corrected sentence is as follows: (line 120 )

By substituting equation 1 and equation 2 into equation 3, the orbit dynamics equation containing the satellite rotation angular velocity is obtained as follows:

  1. Page 4, line 115

It is suggested to add a sentence about the consequences of not assuming the reference satellite in an ideal state.

Answer: The author has added a sentence to the consequences of not assuming the reference satellite in an ideal state in the revised manuscript, and the added sentence is as follows: (line 132)

This will prevent the entire formation satellite from being in an unstable state at the same time

  1. Page 5, line 135

Please rewrite for clarity

Answer: The author has rewritten for clarity in the revised manuscript, and the sentences is as follows: (line 151-154 )

By substituting the above error equations 8 to 14 into the kinematics and dynamics equations 4, 5, 6, and 7, the following attitude and orbit kinematics and dynamics equations for formation satellite errors can be obtained:

  1. Page 5, line 136

Please specify the terms with the suffix "ie". In the text, only the relative terms with the suffix "e" have been described.

Answer: The author has added a note to the suffix "ie" in the revised manuscript, and the added note is as follows: (line 150 )

In the paper, the ‘’ subscript at the lower right of the satellite state parameters is uniformly expressed as the state error parameters of the i th satellite.

  1. Page 11, line 251

It is recommended to specify what is the ideal attitude for the reference satellite (e.g., nadir pointing).

Answer: This paper focuses on verifying the performance of the designed controller. The state of the reference satellite should be random and general. Of course, it is better that the specified ideal state meets the actual requirements. The author has made the modification in the revised manuscript. (line 269-285 )

  1. Page11, line 254 and 255

Please consider specifying how much the attitude of satellite formation should be consistent with the reference satellite.

Answer: The author has added a explanation about how much the attitude of satellite formation should be consistent with the reference satellite. in the revised manuscript, and the explanation e is as follows: (line 276-279)

The attitude angle of the formation satellite and the attitude angle of the reference satellite are kept within the error range of 0.01°. The attitude angular velocity of the formation satellite and the attitude angular velocity of the reference satellite are kept within the error range of 0.01 °/s,

  1. Page16, line 342

It is suggested to clarify how the delay matrix has been derived

Answer: The author has added a explanation about the delay matrix has been derived in the revised manuscript, and the explanation e is as follows: (line 368-369 )

In actual engineering, the delay of the satellite is smaller than that set. The large delay is set to verify the performance of the designed controller.

Reviewer 2 Report

This paper studies the consistency cooperative control of attitude orbit coupling for SAR satellite formations under communication constraints. As the core of the contribution, this paper designs a hierarchical saturated consistency cooperative controller for attitude orbit coupling and uses the Lyapunov direct method to prove the stability of the designed consistent cooperative controller under uncertain space disturbances. The method is expected to have widespread prospects for satellite formations. The manuscript can be published after the following problems are solved.

1.     The introduction part needs to be improved and some latest research is suggested to be added, for instance:

Burnett E R, Schaub H. Geometric perspectives on fundamental solutions in the linearized satellite relative motion problem[J]. Acta Astronautica, 2022, 190: 48-61.

Bai S, Han C, Sun X, et al. Practical maintenance strategies for teardrop hovering formation relative to elliptical orbit[J]. Acta Astronautica, 2022, 190: 176-193.

2,  The simulation results are quite well. It is suggested to compare with other previous method if it is possible, or explain why other control method is not suitable for this problem.

Author Response

Dear Editor,

We are very grateful to the reviewers for their comments, which are extremely valuable to improve the quality of our manuscript. We have considered seriously the suggestions and discussed how to revise the manuscript. We have carefully checked that the references cited in the revised manuscript are related to the content of the manuscript. The revisions are presented in detail in the following. In order to facilitate you to quickly find the modified parts of the manuscript, we use the "track changes" function in MS word. In addition, we also highlight the modified main parts in the revised manuscript. In addition, as the reviewer pointed out that there are many language problems in the manuscript I submitted, the revised manuscript we submitted this time has been provided with professional language editing services, and we will attach relevant certificates at the end.

  1. The introduction part needs to be improved and some latest research is suggested to be added, for instance:

Answer: The author has added some latest research in the revised manuscript, and the latest research are as follows:

SAR satellite formation can achieve high-efficiency ground moving target detection performance through cooperative work and imaging processing of multiple spaceborne radars. When a SAR satellite formation carries out the ground moving target detection task, high-precision configuration position control and attitude tracking control is required. The high-precision control of satellites is conducive to the completion of satellite missions. In some current studies, the high-precision formation satellite flight control is mainly based on the consistency collaborative control [1-11]. Consistency collaborative control means that the movement of the actuator control system tends to be consistent through the exchange of fusion information, continuous feedback and exchange of information among all independent agents in the multiagent system. In the formation flight control of SAR satellites, the relationship between information exchange between SAR satellites and the performance of the formation control system can be obtained through consistency cooperative control. This is beneficial for the design of the formation flight control system and makes the control form of the formation system more general. Reference [6] designed a robust cooperative control law for the hovering target of spacecraft formation by introducing consistent control theory under the condition that the electromagnetic interaction model and dynamic equations are uncertain. Reference [7] proposed an adaptive cooperative collision avoidance control law with strong robustness based on consistency cooperative control and considering various constraints for flight formation. Refer[9] designed a cooperative control law for the optimal orbital manoeuvre of spacecraft based on the first-order consistency control theory for the cooperative flight manoeuvre mission of spacecraft in formation. Reference [10] designed a fault-tolerant controller based on sliding mode cooperative control and showed that the proposed control method tolerates the actuators' faults and controls the satellite's attitude while desaturating the reaction wheels. Reference [11] proposed an coordinated control to fulfill these constraints for impulsive formation maintenance tailored to distributed synthetic aperture radar.

When the SAR satellite formation carries out consistent cooperative control, the member satellites must interact with each other to determine their own control behaviour. However, in the process of information exchange, on the one hand, considering the constraints of distance and signal receiving equipment, and on the other hand, considering factors such as satellite formation manoeuvre, obstacle avoidance, fault and so on, it is inevitable that topology switching, communication delay and other problems will occur. Those factors will reduce the control accuracy of the whole satellite formation control system and affect the stability of the control system. To solve the above problems, Reference [12] designed a terminal sliding mode attitude cooperative controller and proposed a design method based on a spacecraft formation cooperative controller based on an exponential logarithmic sliding mode surface for the situation of communication delay and signal quantization between satellite formations. Reference [13] designed a robust controller with good steady-state performance for the case of communication delay and signal quantization between satellite formations. Reference [14] put forward an optimal control method with guaranteed-performance and switching topologies for the formation achievement problem for swarm systems. Reference [15] designed a distributed model predictive control algorithm considering multiple constraints in order to realize trajectory tracking and formation keeping of multi­UAV system on the premise of meeting the above constraints. In Reference [16], time delay estimation was used to estimate parameter uncertainty and uncertainty caused by external disturbances in satellite dynamics. Then, the time delay estimation output was combined with a robust TSM controller based on interval type-2 fuzzy logic. In Reference [17], an adaptive continuous robust controller was designed to compensate for the influence of model uncertainty on satellite formation when the time delay is uncertain.

The information exchange among the members of the SAR satellite formation depends on wireless communication, while the traditional SAR satellite formation cooperative control technology often ignores some unfavorable factors, such as the communication delay factor. Therefore, this paper studies the cooperative control of SAR satellite formation flight under multiple constraints. Considering the constraints of actuator output capability and uncertain space disturbance, this paper designs a hierarchical saturated consistency coordinated control of attitude orbit coupling in order to solve the problem of communication delay in the configuration for keeping control of SAR satellite formation. In the end, this paper proves the stability of the controller.

2,The simulation results are quite well. It is suggested to compare with other previous method if it is possible, or explain why other control method is not suitable for this problem.

Answer: The author has added explanations about the control method is suitable for this problem.

in the revised manuscript, and the explanations are as follows: (line 178-183,212-213)

(1)When there are reference satellites outside, the member satellites in the formation should not only meet the relative control between member satellites but also meet the absolute control between member satellites and external reference satellites. This requires that the motion between member satellites and member satellites and between member satellites and reference satellites be consistent. This chapter designs the control input for the formation system and the state feedback for the system communication topology.

(2) In order to consider various constraints such as the actuator of the satellite in practice, the hierarchical saturated consistency controller is adopted.

Round 2

Reviewer 2 Report

I think the paper in this stage can be accepted, thank you

Author Response

我们非常感谢审稿人的意见,这些意见对提高我们的手稿质量非常有价值。我们认真考虑了这些建议,并讨论了如何修改手稿。